# Global and Conditional Disruption of the *Igf-I* Gene in Osteoblasts and/or Chondrocytes Unveils Epiphyseal and Metaphyseal Bone-Specific Effects of IGF-I in Bone

**DOI:** 10.3390/biology12091228

**Published:** 2023-09-12

**Authors:** Weirong Xing, Chandrasekhar Kesavan, Sheila Pourteymoor, Subburaman Mohan

**Affiliations:** 1VA Loma Linda Healthcare Systems, Musculoskeletal Disease Center, Loma Linda, CA 92357, USA; weirong.xing@va.gov (W.X.); chandrasekhar.kesavan@va.gov (C.K.); sheila.pourteymoor@va.gov (S.P.); 2Departments of Medicine, Loma Linda University, Loma Linda, CA 92354, USA; 3Departments of Biochemistry, Loma Linda University, Loma Linda, CA 92354, USA; 4Departments of Orthopedic Surgery, Loma Linda University, Loma Linda, CA 92354, USA

**Keywords:** *Igf-I*, knockout, bone, chondrocyte, osteoblast, endochondral ossification, epiphysis, secondary spongiosa, bone mass

## Abstract

**Simple Summary:**

To investigate the relative importance of IGF-I expression in various types of bone cells for endochondral ossification, we examined the trabecular bone phenotypes at the distal femoral epiphysis and the secondary spongiosa of male mice with a global deletion of the *Igf-I* gene, as well as the conditional disruption of *Igf-I* in osteoblasts, chondrocytes, and osteoblasts/chondrocytes and their corresponding control littermates. We demonstrated that the disruption of *Igf-I* globally attenuated bone size much more severely than conditional abrogation in osteoblasts. Trabecular bone mass was lower in the secondary spongiosa of all four knockout mouse lines studied. Global *Igf-I* abrogation, but not conditional loss of *Igf-I*, locally diminished trabecular bone mass in the epiphysis. Our findings provide evidence that local and endocrine IGF-I actions in bone are pleiotropic and dependent on cell type and the bone compartment in which IGF-I acts.

**Abstract:**

To evaluate the relative importance of IGF-I expression in various cell types for endochondral ossification, we quantified the trabecular bone at the secondary spongiosa and epiphysis of the distal femur in 8–12-week-old male mice with a global knockout of the *Igf-I* gene, as well as the conditional deletion of *Igf-I* in osteoblasts, chondrocytes, and osteoblasts/chondrocytes and their corresponding wild-type control littermates. The osteoblast-, chondrocyte-, and osteoblast/chondrocyte-specific *Igf-I* conditional knockout mice were generated by crossing *Igf-I* floxed mice with Cre transgenic mice in which Cre expression is under the control of either the *Col1α2* or *Col2α1* promoter. We found that the global disruption of *Igf-I* resulted in 80% and 70% reductions in bone size, defined as total volume, at the secondary spongiosa and epiphysis of the distal femur, respectively. The abrogation of *Igf-I* in *Col1α2*-producing osteoblasts but not *Col2α1*-producing chondrocytes decreased bone size by 25% at both the secondary spongiosa and epiphysis. In comparison, the deletion of the *Igf-I* globally or specifically in osteoblasts or chondrocytes reduced trabecular bone mass by 25%. In contrast, the universal deletion of *Igf-I* in all cells, but not the conditional disruption of *Igf-I* in osteoblasts and/or chondrocytes reduced trabecular bone mass in the epiphysis. The reduced trabecular bone mass at the secondary spongiosa in osteoblast- and/or chondrocyte-specific *Igf-I* conditional knockout mice is caused by the reduced trabecular number and increased trabecular separation. Immunohistochemistry studies found that the expression levels of chondrocyte (COL10, MMP13) and osteoblast (BSP) markers were less in the secondary spongiosa and the epiphyses in the global *Igf-I* deletion mice. Our data indicate that local and endocrine *Igf-I* act pleiotropically and in a cell type- and bone compartment-dependent manner in bone.

## 1. Introduction

Bone size and bone mineral density (BMD) determine bone strength. Several potential regulatory molecules contribute to skeletal changes during post-natal growth. One of them is the insulin-like growth factor 1 (*Igf-I*), which has received considerable attention for several reasons. Firstly, mice with targeted disruption of the *Igf-I* gene had compromised bone size and BMD [1]. Total BMD, femoral cortical BMD, and femur bone length were reduced by 68%, 29%, and 42% in the global *Igf-I* gene knockout (KO) mice at 8 weeks of age [1]. The periosteal circumference of the femur was reduced by 46% compared to the control wild-type (WT) mice. The deletion of *Igf-I* completely blunted the periosteal expansion during puberty. Secondly, targeted overexpression of *Igf-I* in osteoblasts increased peak BMD caused by the increased activity of resident osteoblasts [2]. Femoral trabecular and cortical BMD were increased by 10% and 4% in osteoblast-specific *Igf-I* transgenic mice at 6 weeks of age. Femoral bone volume to total volume was increased by 28% [2]. Treating adult ovariectomized rats with IGF-1 increased trabecular bone mass in the distal femoral metaphysis, epiphysis, and lumbar vertebral body [3].

Regarding the relevance of these findings in explaining peak BMD variation in humans, we have shown that the serum level of IGF-I increases during puberty and correlates with bone size and BMD [4]. Furthermore, the findings that both the variation in peak BMD and circulating levels of IGF-I are largely genetically determined provide evidence that the differences in *Igf-I* expression caused by gene polymorphism could, in part, contribute to peak BMD differences and, therefore, the risk of osteoporosis [5]. In terms of mechanisms of IGF-1 regulation of bone size and peak BMD, both endocrine and local autocrine/paracrine actions of IGF-I have been proposed [6,7]. Much of the circulating IGF-I is known to be primarily produced by liver hepatocytes, which enter the blood circulation and act as an endocrine hormone [6]. In mice with a liver-specific abrogation of *Igf-I*, the circulating IGF-I protein level was reduced by more than 75% compared to normal levels [8,9]. Despite the great reduction in the systemic level of IGF-I, hepatic *Igf-I* conditional KO mice grew normally [8]. The appendicular skeletal growth of the liver-specific *Igf-I* conditional KO mice, as determined based on body weight, body length, and femoral length, did not differ from that of wild-type littermates [8]. However, the adult axial skeletal growth and the cortical bone width were reduced in the liver-specific conditional KO mice [10]. In contrast, the global deletion of the *Igf-I* gene in every cell caused a 20–40% reduction in femur length, size, and BMD [1]. Our studies and those of other researchers strongly suggest that IGF-I, locally produced by the cells that reside in bone, acts in an autocrine/paracrine manner and is sufficient to support skeletal development and growth during puberty [11,12]. However, the relative contribution of the IGF-I produced by specific skeletal cell types to skeletal development remains unclear.

Previous studies have demonstrated that the augmentation of thyroid hormone levels during the pre-pubertal growth period is essential for endochondral ossification, which occurs at the epiphyses and secondary spongiosa of the tibias and femurs. Thyroid hormone stimulation of endochondral bone formation is predicted to be mediated via the activation of several growth factor signaling pathways. One such mediator is IGF-I, which elicits endocrine and local actions in different bone cell types [13]. To determine the relative importance of different sources of IGF-I in mediating skeletal growth, we carried out microCT scanning to evaluate the trabecular bone phenotypes at the distal femoral epiphysis and secondary spongiosa of mice with global abrogation of the *Igf-I* gene, as well as the conditional disruption of the *Igf-1* in osteoblasts and/or chondrocytes.

## 2. Materials and Methods

### 2.1. Generation of Igf-I KO Mice

The generation of *Igf-I* global and conditional KO mice is illustrated in Figure 1. The global *Igf-I* KO mice in which the exon 4 of *Igf-I* gene is disrupted in every cell in the body were generated as reported previously [1]. The osteoblast-specific *Igf-I* conditional KO mice in which the exon 4 of *Igf-I* gene is only deleted in type I collagen-producing osteoblasts were generated by crossing *Igf-I* floxed mice with *Col1α2*-Cre mice, as described in [14]. The chondrocyte-specific *Igf-I* conditional KO mice in which the exon 4 of the *Igf-I* gene is only disrupted in type II collagen-expressing chondrocytes were produced by breeding *Igf-I* floxed mice with *Col2α1*-Cre mice [15]. The osteoblast- and chondrocyte-specific *Igf-I* double conditional KO mice in which the exon 4 of *Igf-I* is abrogated in both cell types were produced by crossing *Igf-I* floxed mice with Cre double transgenic mice, in which Cre expression is driven by the regulatory elements of the *Col1α2* and *Col2α1* genes [14,15]. Cre-negative homozygous floxed mice were used as control WT mice. Cre-positive homozygous floxed mice were considered as conditional KO mice. In our previous studies, we have demonstrated that the osteoblast-specific *Igf-1* conditional knockout (cKO) mice generated using the A26 line of *Col1α1*-Cre also generated conditional mutants with normal skeletal structures at the expected ratio [14]. *Col1α2*-Cre expression was primarily localized in osteoblasts in the tibia of newborn conditional mutants but not in WT control mice. The Cre expression was not detectable in the liver of either the conditional mutant or control WT littermate mice. The expression of *Igf-I* in the long bones of osteoblast-specific conditional mutants was attenuated by 70% compared to control mice [14]. The chondrocyte-specific *Igf-1* cKO mice directed under the control of the *Col2α1* promoter were normally born at the expected ratio of 50% conditional mutant and 50% control mice. No significant differences in skeletal parameters were found at 2 weeks after birth. Cre expression was primarily detected in chondrocytes but not in primary osteoblasts. IGF-I expression was diminished by 40% in the long bones but undetectable in non-bone tissues of chondrocyte-specific *Igf-I* cKO mice [15]. No differences in the circulating serum level of IGF-I were found in either osteoblast- or chondrocyte-specific *Igf-1* cKO mice compared to the corresponding control mice [14,15]. Mice were housed in the Loma Linda VA Healthcare System (LLVAHS) with controlled temperature (22 °C) and illumination (14 h of light and 10 h of dark) and unrestricted food and water. All procedures were carried out via a protocol (MOH0029/204) approved by the Institutional Animal Care and Use Committee of the LLVAHS. Mice were anesthetized with isoflurane prior to ear punching and tail clipping, which was performed to enable genotyping. Experimental mice were euthanized via overexposure CO_2_, followed by decapitation.

### 2.2. MicroCT Evaluation

Mouse axial skeleton length were measured after euthanasia. The femur length was measured after the dissection of the femur and removal of the soft tissues prior to microCT scanning. The epiphysis and secondary spongiosa of the femurs of 12-week-old global *Igf-I* KO and 8-week-old osteoblast- and/or chondrocyte-specific *Igf-I* conditional KO male mice and their WT gender-matched littermates were scanned via X-ray at 55 kVp with a voxel size of 10.5 µm using the vivaCT 40 microCT system acquired from Scanco (Scanco Medical, Bruttisellen, Switzerland). The trabecular bone of the epiphysis was scanned from the top to the bottom of the femoral epiphysis. The femoral trabecular bone of the secondary spongiosa region started at 0.36 mm from the distal growth plate in the direction of the metaphysis and extended for 180 slices (1.89 mm) for WT control mice. As the bone length in *Igf-I* KO mice was significantly changed, the location of slices selected for use in analyses was adjusted for bone length, ensuring that the analyzed regions of the bone samples were anatomically comparable. The starting point for analysis was calculated via the following formula: bone length of mutant mouse/mean bone length of WTmice × 0.36 mm. The number of slices analyzed was calculated via the following formula: bone length of mutant mouse/mean bone length of WTmice ×180 slices. Based on the calculation, the range of the number of slices analyzed was 110–180 for global KO and conditional KO mice. The average number of slices was 110, 166, 180, and 155 for global, osteoblast-specific, chondrocyte-specific, and osteoblast/chondrocyte-specific conditional KO mice, respectively. The total volume (TV, mm^3^), bone volume (BV, mm^3^), Bone volume fraction (BV/TV, %), trabecular number (Tb. N, mm^−1^), trabecular thickness (Tb. Th, mm), and trabecular separation (Tb. Sp, mm) were evaluated as reported [16]. The cross-sectional area (CSA) of the femoral metaphyseal secondary spongiosa was calculated by dividing the TV by the scan length (10.5 µm × slice number).

### 2.3. Immunohistochemistry

Frozen bone sections were prepared via a cryostat and fixed in 4% paraformaldehyde at 4 °C for 3 days. After 3-day decalcification in 14% EDTA at 4 °C under constant agitation and washing with PBS, the bone sections were soaked in 30% sucrose in PBS at 4 °C overnight. The slides were pre-treated with a blocking solution containing normal goat serum for 20 min and incubated with primary antibodies at dilutions of 1:100 for COL10 (ab58632, Abcam, Waltham, MA, USA), MMP13 (nbp1-45723, Novus, St. Louis, MO, USA), and BSPII (a kind gift from Dr. Renny Franceschi, University of Michigan) for 30 min at room temperature. After 3 times of washing with PBS, the sections were incubated with the secondary anti-mouse or anti-rabbit Dylight 488-fluorochrome labeled antibody at 1x pre-dilution (Vector Laboratories, Burlingame, CA, USA) for another 30 min at room temperature. The sections were again washed with PBS and mounted using Vectashield mounting medium with DAPI (Vector Laboratories, Inc., Burlingame, CA, USA).

### 2.4. Statistical Analysis

Data were analyzed via Student’s *t*-test. Values are presented as mean ± SEM (*n* = 6–9 male mice per genotype).

## 3. Results

### 3.1. Bone Size at the Epiphysis Is Reduced in Mice with Global and Osteoblastic-Specific Disruption of the Igf-I Gene but Not in Chondrocyte-Specific Conditional KO Mice

Body weight was reduced by 77%, 34%, 18%, and 35% in the global *Igf-I* KO, osteoblast-specific, chondrocyte-specific, and osteoblast/chondrocyte-specific *Igf-I* conditional KO mice, respectively, compared to the control WT mice (Figure 2A–D). Body length was also diminished by 40%, 12%, and 17% in the global *Igf-I* KO, osteoblast-specific, and osteoblast/chondrocyte-specific *Igf-I* KO mice, respectively. Body length was not significantly different between the chondrocyte-specific conditional *Igf-I* conditional KO and control mice (Figure 2C). In line with the reduced body length and femur length, the CSA of the femoral metaphyseal secondary spongiosa was also significantly decreased in the global *Igf-I* KO, osteoblast-specific, and osteoblast/chondrocyte-specific *Igf-I* KO mice (Figure 2A,B,D). The CSA of the femoral metaphyseal region was reduced by 68%, 29%, and 33% in the global, osteoblast-specific, and osteoblast/chondrocyte-specific *Igf-I* KO mice, respectively.

MicroCT analyses revealed that disruption of the *Igf-I* gene in every cell type in the global *Igf-I* KO mice resulted in a 70% reduction in bone size, which is defined as the total volume at the epiphysis of the distal femur. Trabecular bone volume adjusted to total volume (BV/TV) and volumetric BMD (vBMD) were reduced by 25% and 40%, respectively, at the epiphysis of the distal femur compared to WT control mice. The reduced bone volume and vBMD occurred due to a significant reduction in trabecular thickness (Figure 3A,B). Mice with a loss of *Igf-I* in osteoblasts but not in chondrocytes showed a 25% decrease in bone size at the epiphysis. Compared to control littermate siblings, mice with the conditional deletion of the *Igf-I* gene in osteoblasts and chondrocytes exhibited a 25% decrease in bone size at the epiphysis of the femur (Figure 3B).

### 3.2. Trabecular Bone Volume and vBMD Are Reduced at the Secondary Spongiosa of the Distal Femur in Mice with the Disruption of the Igf-I Gene

The disruption of the *Igf-I* gene in every cell type exhibited an 80% reduction in bone width, as reflected by the TV. By comparison, mice with the deletion of the *Igf-I* gene either in osteoblasts or both osteoblasts and chondrocytes displayed a 25% reduction in bone size at the secondary spongiosa of the distal femur (Figure 4A,B). In contrast, no change in bone size was found at the secondary spongiosa of the distal femur in mice with the disruption of the *Igf-I* in chondrocytes. Trabecular BV/TV and vBMD were also 25% and 40%, respectively, lower in the global *Igf-I* KO mice compared to the control littermates. The specific disruption of the *Igf-I* gene in osteoblasts, chondrocytes, or both osteoblasts and chondrocytes reduced trabecular BV/TV and vBMD by 25% (Figure 3B). The reduced trabecular bone mass in the global and osteoblast- and/or chondrocyte-specific *Igf-I* conditional KO mice was primarily caused by a reduced trabecular number and increased trabecular separation. The trabecular thickness was reduced in the secondary spongiosa of the distal femur in the global *Igf-I* KO mice but not in the osteoblast-, chondrocyte- or osteoblast/chondrocyte-specific conditional KO mice. Immunofluorescent staining found that collagen 10 (COL10) and matrix metallopeptidase 13 (MMP13), which are two markers of differentiating chondrocytes, but not bone sialoprotein II (BSPII), which is a marker of a differentiating osteoblast, were expressed in growth plate chondrocytes, as expected (Figure 5A–D). The expression of BSPII was restricted in the trabecular bone of the secondary spongiosa of the distal femur (Figure 5E,F). In accordance with the micro-CT findings, both COL10- and MMP13-expressing chondrocytes and BSPII-expressing osteoblasts were markedly reduced in the secondary spongiosa of the *Igf-I* KO males compared to control mice (Figure 5A–F).

## 4. Discussion

Although IGF-I is known to be produced by many cell types in the body, hepatocytes are the primary contributor to the circulating levels of IGF-I [8,9]. The IGF-I produced in the liver acts in an endocrine manner, primarily circulating in the blood as a ternary complex with an acid labile subunit (ALS) and an IGF-binding protein-3 [17]. Growth hormone and thyroid hormones are major regulators of IGF-I expression in hepatocytes [18,19]. Growth hormone deficiency in childhood decreased BMD, and growth hormone replacement in these children increased bone growth and bone strength [20]. A positive correlation between the serum IGF-I and BMD exists in humans [21,22]. Lower levels of serum IGF-I in women are associated with increased osteoporotic fractures [23]. Local IGF-I expression is induced by systemic hormones, such as the growth hormone and thyroid hormone, as well as local growth factors, including BMP-7 and TGFβ1, to act in an autocrine/paracrine manner [13,18,24]. Both endocrine and local IGF-I actions have been implicated in promoting skeletal growth [1,15]. In terms of the relative roles played by circulating and locally produced IGF-I in regulating bone growth and trabecular bone mass, previous studies have shown that the liver-specific KO of *Igf-I* in mice caused a more than 80% reduction in circulating IGF-I levels and an increase in growth hormone. Nonetheless, these mice developed normally [8]. The circulating IGF-I level was further reduced in mice lacking both liver-derived IGF-I and ALS, causing a reduction in bone size [1,25]. The reduction in bone size in *Igf-I* and *Als* double KO mice was much smaller than that of the global *Igf-I* KO mice, suggesting that locally produced IGF-I plays a pivotal role in promoting normal bone growth during development. However, the importance of IGF-I produced by different cell types and tissue compartments in bone in mediating skeletal growth has not been fully elucidated.

In this study, we hypothesized that circulating IGF-I and locally produced IGF-I in various cell types differentially contribute to endochondral ossification between the metaphysis (primary ossification center) and the epiphysis (secondary ossification center) of the long bones. While both primary and secondary ossification centers are formed via endochondral ossification, important differences exist between them, including the time they occur [26]. In previous studies, we found that the thyroid hormone is essential for the initiation and progression of the secondary ossification center at the epiphysis. Since the thyroid hormone has been shown to stimulate *Igf-I* expression [13], we evaluated if trabecular bone formation at the epiphysis is also IGF-I dependent, as in the case of metaphysis. To study the different role played by IGF-I in endochondral bone formation, we evaluated the phenotypes of the trabecular bone at the femoral epiphysis and the secondary spongiosa of males with a global disruption of the *Igf-I* gene, as well as the conditional abrogation of the *Igf-I* in osteoblasts and/or chondrocytes, and their corresponding control littermates. We only analyzed males because female mice have estrous cycles that cause a significant variation in bone growth and remodeling. Consistent with a previous study [25], global *Igf-I* KO mice exhibited a more significant reduction in bone size, as evidenced by their reduced body weight, body length, femur length, and femur cross-sectional area, compared to osteoblast and chondrocyte double conditional KO mice. We observed that the disruption of the *Igf-I* gene in every cell type in the global IGF-I KO mice resulted in 80% and 70% reductions in bone size at the secondary spongiosa and the epiphysis of the distal femur, respectively. Deleting the *Igf-I* gene in type I collagen-producing osteoblasts but not type II collagen-producing chondrocytes in mice decreased bone size by 25% at both the secondary spongiosa and the epiphysis of the femur. The loss of the *Igf-I* gene globally or specifically in osteoblasts or chondrocytes attenuated trabecular mass by 25% due to the reduced trabecular number and increased separation. Trabecular bone was thinner in both the secondary spongiosa, and the epiphysis of global *Igf-I* KO mice but not in the *Igf-I* conditional KO mice. In congruence with the findings obtained via micro-CT scanning, the differentiation of both osteoblasts and chondrocytes was severely compromised, as evidenced by the reduced expression of COL10 and MMP13 in the chondrocytes of the secondary spongiosa and diminished BSPII staining in the distal metaphysis of the femur in global *Igf-I* KO mice.

Collagen 2a1 expression is localized in the cartilaginous primary spongiosa but not in the secondary spongiosa, where the trabecular bone volume and BMD were significantly reduced in the mice with a deficiency of IGF-I in chondrocytes. The issue to consider is, thus, how the disruption of chondrocyte-produced IGF-I affects trabecular bone formation in the secondary spongiosa region of the femur. A previous study found that growth plate hypertrophic chondrocytes can transdifferentiate to produce osteoblasts at the primary spongiosa region [26]. Therefore, trabecular bone changes in the secondary spongiosa may reflect the direct effects of *Igf-I* gene disruption in chondrocytes, influencing chondrocyte differentiation into osteoblasts and bone formation at the primary spongiosa. Our data support the idea of a larger role being played by local IGF-I than endocrine IGF-I in stimulating bone growth and development. Locally, osteoblast-derived IGF-I but not chondrocyte-derived IGF-I promotes bone growth. However, osteoblast-derived IGF-I and chondrocyte-derived IGF-I play the same important role in regulating trabecular bone density at the secondary spongiosa but not at the epiphysis. Interestingly, the deletion of *Igf-I* globally but not locally in bone cells in mice compromises trabecular bone mass at the epiphysis. Our studies provide evidence that circulating IGF-I is a major contributor to epiphysis ossification. Thus, the universal and conditional disruption of the *Igf-1* gene in osteoblasts and/or chondrocytes in mice leads to the discovery of cell type- and tissue compartment-specific effects of IGF-I in bone, as summarized in Table 1.

Other studies have evaluated the role played by IGF-I expressed in various bone cell types in regulating skeletal growth, repair, and remodeling [27]. IGF-I expressed in osteoblasts or osteocytes, but not liver hepatocytes, is indispensable for performing mechanical loading-induced bone formation [28]. While osteocyte-derived IGF-I plays an important role in skeletal development, surprisingly, the disruption of the *Igf1* gene in osteocytes did not impede but promoted fracture callus remodeling, as well as bone repletion response, in mice [29,30]. Our previous study found that the disruption of the *Igf1* gene in chondrocytes led to a significant reduction in cortical bone size measured via peripheral quantitative computed tomography at the mid-diaphysis of the femur [15]. In contrast, there was no change in the CSA as measured via micro-CT at the epiphysis of chondrocyte- specific *Igf1* conditional KO mice in the present study. These data are consistent with the complex roles played by IGF-I produced by different bone cell types in regulating bone metabolism.

The mechanism through which IGF-I regulates cell- and compartment-specific actions in bone is unknown. While we did not determine the expression level of the IGF-I receptor in osteoblasts, chondrocytes, and other tissues in this study, the IGF-I receptor is widely expressed in many tissues in the body. Whether the disruption of *Igf-1* expression in osteoblasts and/or chondrocytes produces a compensatory increase in IGF-I receptor expression remains to be determined. Besides systemic hormones and local growth factors, mechanical strain is an important regulator of *Igf-1* expression in bone cells [30]. Moreover, IGF-I’s actions are controlled by IGF-binding proteins and their proteases. Thus, the local actions of IGF-I in bone are likely to be subject to regulation by a variety of signals to meet the demands of the growing skeleton. In addition, the IGF-I expression in local osteoprogenitors, and other cells in the bone marrow microenvironment may contribute to local bone cell proliferation, differentiation, and skeletal growth.

The limitations of this study are as follows: (1) the failure to include female gender to confirm the cell-type and compartment-specific effects of IGF-I in female mice; (2) the lack of cortical bone analyses to determine the role played by IGF-I expressed in chondrocytes and osteoblasts in cortical bone volume regulation; (3) the quantitative parameters of the cartilage and chondrocytes, IGF-I receptor expression, IGF-I receptor phosphorylation/activation and downstream signaling in the growth plate and primary/secondary spongiosa were not provided in this study.

## 5. Conclusions

In the present study, we compared the trabecular bone phenotypes at the secondary spongiosa and the epiphysis of the distal femur in mice to the global disruption of the *Igf-I* gene and the conditional deletion of the *Igf-I* gene in the osteoblasts, chondrocytes, osteoblasts/chondrocytes, and control littermates. We found that the global disruption of *Igf-I* caused a 70–80% attenuation in bone size at the secondary spongiosa and the epiphysis of the distal femur. The disruption of *Igf-I* in osteoblasts but not chondrocytes resulted in a 25% smaller bone size at the secondary spongiosa and epiphysis. In comparison, the deletion of the *Igf-I* globally or specifically in osteoblasts or chondrocytes leads to a 25% decrease in trabecular bone mass. In contrast, the deletion of *Igf-I* in all cells, but not the conditional disruption of *Igf-I* in osteoblasts and/or chondrocytes, reduced trabecular bone mass during epiphysis due to reduced trabecular number and increased trabecular separation. Our data suggest that local and endocrine IGF-I act pleiotropically and in a cell type- and tissue compartment-dependent manner in bone. Our future research directions will be focused on how the loss of *Igf-I* expression in osteoblasts and/or chondrocytes at various bone cells and skeletal sites influences IGF-I receptor expression and downstream signaling pathways in the growth plate and trabecular bone (primary and secondary spongiosa) to provide mechanistic insights into site-specific effects of IGF-I in bone.

## Figures and Tables

**Figure 1 biology-12-01228-f001:**
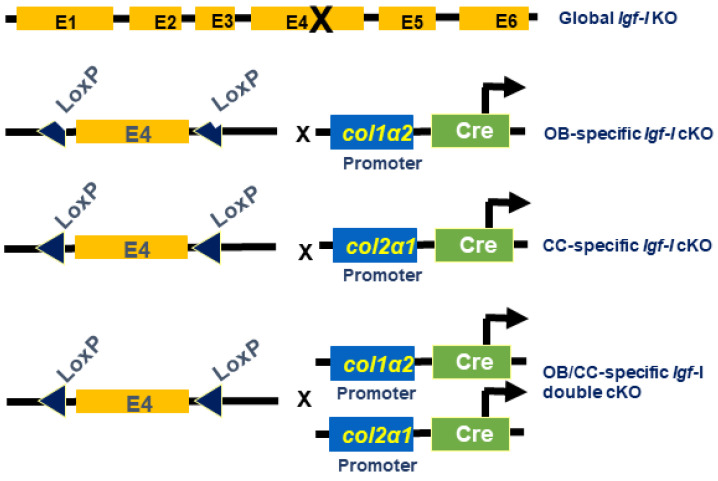
Generation of global knockout (KO) and conditional KO (cKO) of *Igf-I* gene in osteoblasts, chondrocytes, and osteoblasts/chondrocytes. OB, osteoblast; CC, chondrocyte; OB/CC, osteoblast/chondrocyte.

**Figure 2 biology-12-01228-f002:**
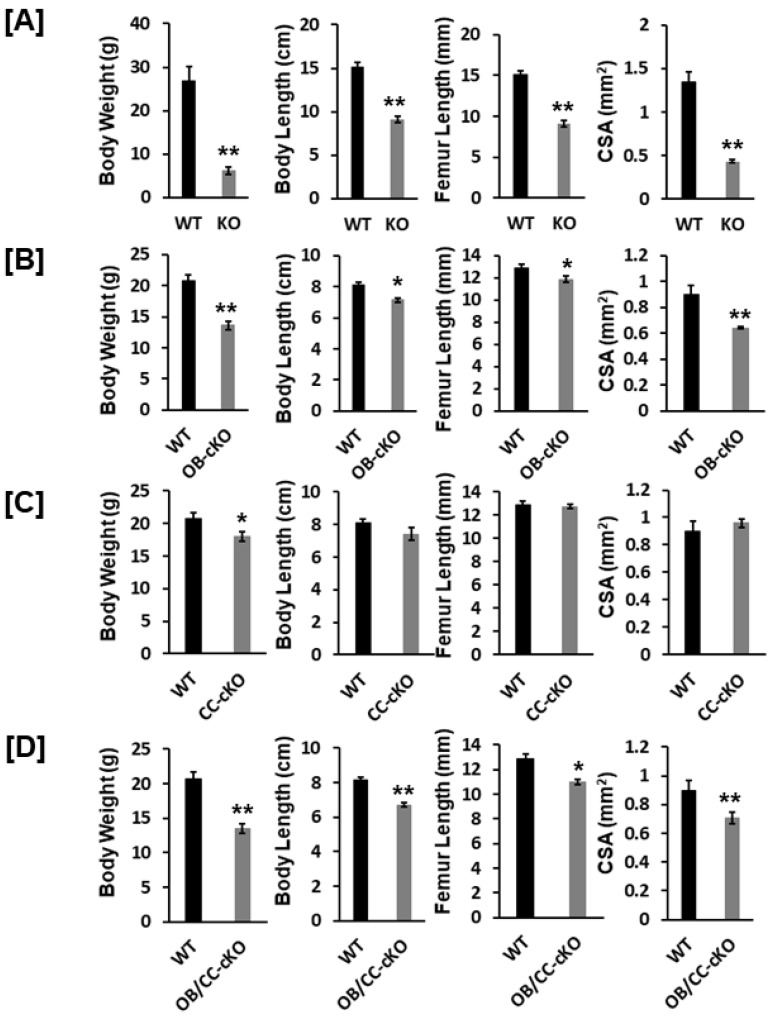
Body weight and bone size were reduced in the global *Igf-I* and osteoblast-specific or osteoblast/chondrocyte-specific conditional KO mice. (**A**) Global KO mice. WT, wild type; KO, knockout (12-week-old mice); (**B**–**D**) Condittional KO mice in osteoblasts, chondrocytes, and osteoblasts/chondrocytes, respectively. cKO, conditional KO (8-week-old mice); OB, osteoblast; CC, chondrocyte; CSA, cross sectional area. Values are mean ± SEM (*n* = 6–9, males). A star (*) indicates *p* < 0.05. Stars (**) indicate *p* < 0.01.

**Figure 3 biology-12-01228-f003:**
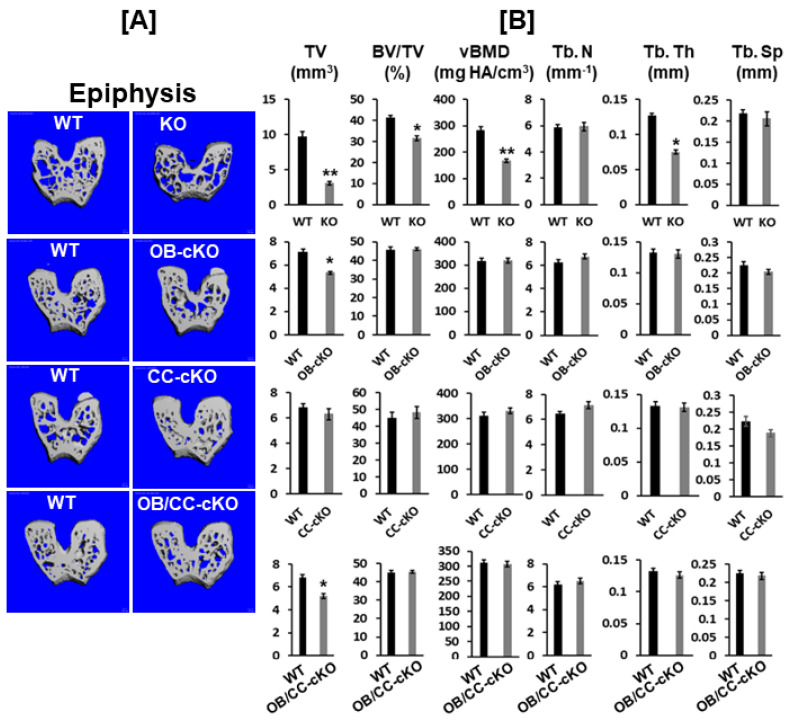
Bone size at the epiphysis is reduced in mice with global and osteoblastic-specific conditional KO of *Igf-I* gene but not in chondrocyte-specific conditional KO mice. (**A**) Micro-CT images of the trabecular bone of the epiphyses of the KO (12-week-old mice) and cKO mice (8-week-old mice). (**B**) Quantitative microCT data of the trabecular bone of the epiphysis in Figure 2A. TV, total volume; BV, bone volume; Tb. N, trabecular number; Tb. Th, trabecular thickness; Tb. Sp, trabecular spacing; vBMD, volumetric bone mineral density. Values are mean ± SEM (*n* = 6–9, males). A star (*) indicates *p* < 0.05. Two stars (**) indicate *p* < 0.01.

**Figure 4 biology-12-01228-f004:**
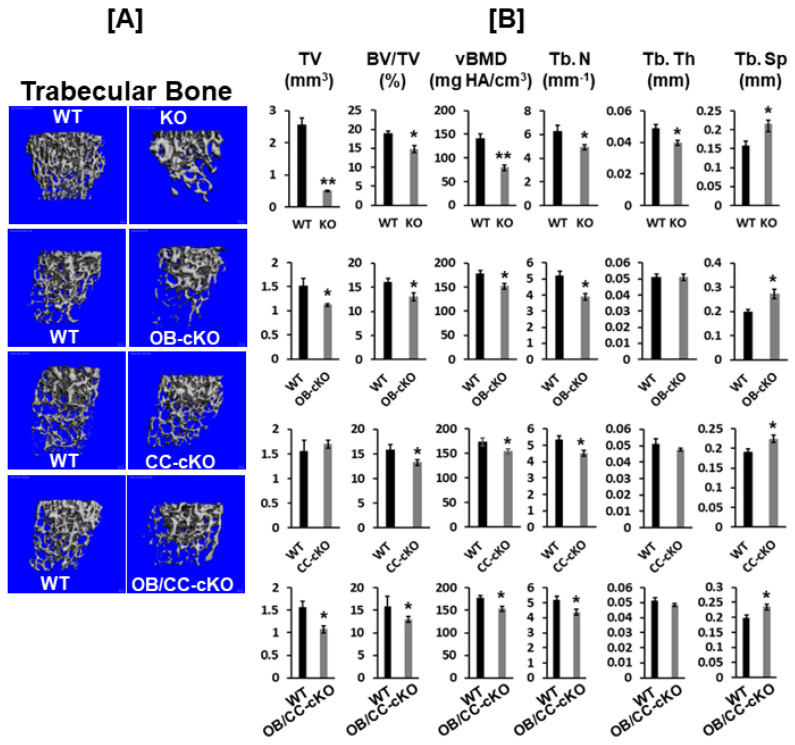
Trabecular bone volume and vBMD are reduced at the secondary spongiosa of the distal femur in mice with the disruption of the *Igf-I* gene. (**A**) MicroCT images of the trabecular bone of the secondary spongiosa of the KO (12-week-old mice) and cKO (8-week-old mice) mice. (**B**) Quantitative micro-CT data of the trabecular bone of the distal femur in Figure 2A. Values are mean ± SEM (*n* = 6–9, males). A star (*) indicates *p* < 0.05. Two stars (**) indicate *p* < 0.01.

**Figure 5 biology-12-01228-f005:**
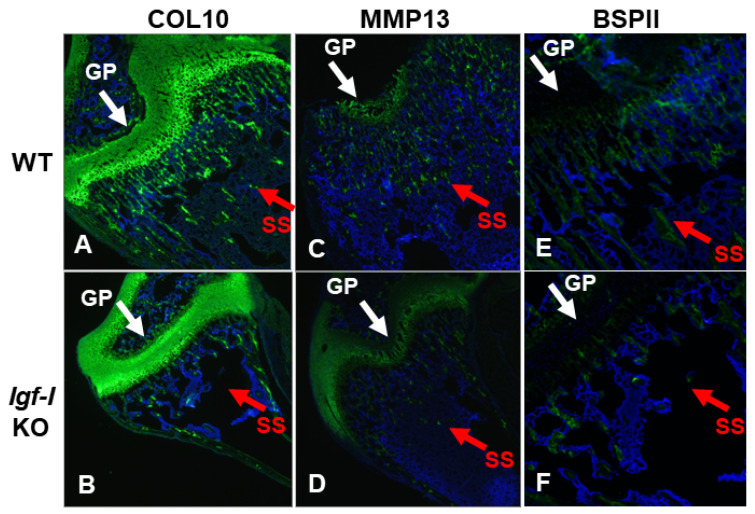
Expression of COL10, MMP13, and BSPII was reduced in distal femur of global *Igf-I* KO. Expression levels of collagen type 10 (COL10), matrix metallopeptidase 13 (MMP13), and bone sialoprotein II (BSPII) were analyzed via immunofluorescent staining. Nuclei were stained in blue with DAPI. Signals are shown in green. (**A**,**C**,**E**) represent longitudinal sections of distal femur from wild-type mice, while (**B**,**D**,**F**) represent longitudinal sections of distal femur from global *Igf-I* KO mice. GP, growth plate; SS: secondary spongiosa. Arrows indicate differentiating chondrocytes expressing COL10 and MMP13 or osteoblasts expressing BSPII (Green).

**Table 1 biology-12-01228-t001:** Summary of bone phenotypes of mice with deletion of *Igf-I* gene.

KO Cell Type	Mechanism	Bone Size	Trabecular Bone
Every cell type (global)	Endocrine and local action	Reduced	Reduced
Osteoblasts	Local action	Reduced	Reduced
Chondrocytes	Local action	No change	Reduced
Osteoblasts/chondrocytes	Local action	Reduced	Reduced
Hepatocytes	Local action	Reduced	No change

## Data Availability

The raw datasets generated and/or analyzed during the current study are available from the corresponding author upon reasonable request.

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
