# Peer review of "Global and Conditional Disruption of the Igf-I Gene in Osteoblasts and/or Chondrocytes Unveils Epiphyseal and Metaphyseal Bone-Specific Effects of IGF-I in Bone"

_biology, 2023, doi:10.3390/biology12091228_

Round 1

Reviewer 1 Report

This manuscript evaluate the relative importance of locally expressed IGF-I on the bone development. The presented data showed that compared to global knockout IGF-I, specific deletion of IGF-I in each of bone cells, osteoblast, chondrocyte and osteoblast/chondrocyte, had less effect on bone development parameters, femoral length and cross-sectional area (CSA), and bone size at the epiphysis. These findings are useful to understand pleiotropic effect of local and endocrine IGF-I on bone development.  The manuscript should provide evidences showing the mechanism to interpret the effects of locally expressed vs. global IGF-I on bone development. The growth plate chondrocyte proliferation and the subsequent endochondral ossification are responsible for the growth of the long bone and regeneration of trabecular bone. The quantitative parameters of cartilage/chondrocyte, IGF-I receptor expression and critical downstream signaling in the growth plate and primary/secondary spongiosa should be provided to determine if partial impairment of these parameters can interpret the phenotype in the conditional knockout mice. In addition, serum IGF-I levels in each knockout mice should be provided.

Other concerns

The body weight, body length and femur length in WT control mice for global knockout mice were higher than those WT mice for OB, CC and OB-CC conditional knockout mice in Fig.2. Similarly, the bone size at the epiphysis in WT mice for global KO was bigger than the WT mice for cKO in Fig.3. What reasons caused the variation of these parameters in the control WT mice?

Author Response

  1. The manuscript should provide evidence showing the mechanism to interpret the effects of locally expressed vs. global IGF-I on bone development. The growth plate chondrocyte proliferation and the subsequent endochondral ossification are responsible for the growth of the long bone and regeneration of trabecular bone. The quantitative parameters of cartilage/chondrocyte, IGF-I receptor expression and critical downstream signaling in the growth plate and primary/secondary spongiosa should be provided to determine if partial impairment of these parameters can interpret the phenotype in the conditional knockout mice. In addition, serum IGF-I levels in each line of knockout mice should be provided.

Response: Serum IGF-I levels in the osteoblast and chondrocyte-specific conditional knockout mice have been previously published and the papers have been cited (Cited references 14 & 15,  Govoni KE, 2007).  No difference in circulating serum level of IGF-I was found in either osteoblast- or chondrocyte-specific Igf-1 cKO mice compared to corresponding control mice. We agree that additional studies on how loss of IGF-I expression in osteoblasts and/or chondrocytes at various skeletal sites influence IGF-I receptor expression and down stream signaling pathways in the growth plate and trabecular bone (primary and secondary spongiosa) could provide mechanistic insights on site-specific effects of IGF-I in bone.  These studies require generation of additional conditional knockout and control mice that involve considerable time and efforts and are beyond the scope of the current study. These will be our future studies. We have added this information in the conclusion in the revised manuscript.

  1. The body weight, body length and femur length in WT control mice for global knockout mice were higher than those WT mice for OB, CC and OB-CC conditional knockout mice in Fig.2. Similarly, the bone size at the epiphysis in WT mice for global KO was bigger than the WT mice for cKO in Fig.3. What reasons caused the variation of these parameters in the control WT mice?

Response: We apologize for not clarifying the age difference in these knockout mice. The control WT littermates for global knockout mice were 12 weeks old. But the control WT littermates for the conditional knockout mice were 8 weeks old. We have indicated the ages in the materials and methods as well as in the figure legends.

Reviewer 2 Report

In the present work the authors aim to evaluate the relative importance of IGF-I expression in various cell types for endochondral ossification. To this end, they perform microCT analysis of the trabecular bone at the secondary spongiosa and epiphysis of the distal femur in adult male mice with global knockout of the Igf-I gene or conditional deletion of this gene in osteoblasts, chondrocytes, osteoblasts/chondrocytes. They obtain different results in the various models used in the study, thus they conclude that the effect exerted by IGF-I depend on the specific bone compartment considered. We think they have to accompany and complete their findings at least with some molecular investigation of downstream pathways.

Please also complete the methods section regarding sample preparation for immunohistochemistry (fixation, decalcification).

Author Response

  1. We think they have to accompany and complete their findings at least with some molecular investigation of downstream pathways.

Response: The molecular investigation of downstream signaling pathways in osteoblast and/or chondrocyte specific IGF-I conditional knockout mice require generation of additional conditional knockout and control mice that involve considerable time and efforts and are beyond the scope of the current study. However, we have added this information as one of the limitations of this study as well as our future directions in the revised manuscript.

  1. Please also complete the methods section regarding sample preparation for immunohistochemistry (fixation, decalcification).

Response:  We apologize for this missing information which is now added in the revised manuscript. We added “Frozen bone sections were prepared via a cryostat and fixed in 4% paraformaldehyde at 4 0C for 3 days. After 3-day decalcification in 14% EDTA at 4 0C under constant agitation and washing with PBS, the bone sections were soaked in 30% sucrose in PBS at 4 0C overnight.”

Reviewer 3 Report

To determine the role of Igf-1 expressed in different cell populations in bone formation, Xing and co-workers compared the skeletal phenotype of mice lacking Igf-1 in different cells populations including osteoblasts and chondrocytes to that of the Igf-1 global knockout mice. The authors found that deletion of Igf-1 from osteoblasts or chondrocytes recapitulated the skeletal phenotype of Igf-1 global knockout mice at metaphysis. In addition, deletion of Igf-1 from both osteoblasts and chondrocytes did not result in a more profound phenotype in this site. However, in epiphysis, deletion of Igf-1 in neither osteoblasts nor chondrocytes produced the skeletal phenotype observed in the global knockout mice.  The authors concluded that “local and endocrine IGF-I actions in bone are pleiotropic and dependent on cell type as well as the bone compartment where IGF-I acts”. This study is descriptive and additional evidence is needed to support the idea that circulating IGF-I is a major contributor of the epiphysis ossification.

Specific concerns:

1.       Is the expression of Igf-1 in osteoblasts different in metaphysis versus epiphysis? How about the relative expression level of Igf-1 in osteoblasts compared to other cell types in epiphysis? In situ measurement of Igf-1 mRNA abundance in different cell populations in epiphysis is needed to determine whether local Igf-1 expression in other cell types compensate the loss of osteoblastic Igf-1 in this skeletal site.

2.       In discussion, please add a paragraph discussing the possibility of the local contribution of Igf-1 expression in progenitor cells within the bone microenvironment to bone formation.  Deletion of Igf-1 using a Cre deleter strain targeting early progenitor cells such as Prx1-Cre may recapitulate the full phenotype of the global knockout mice.

3.       Please check the references. Reference 30, 31 are not correctly cited and references 32-36 are missing.

Author Response

Specific concerns:

  1. Is the expression of Igf-1 in osteoblasts different in metaphysis versus epiphysis? How about the relative expression level of Igf-1 in osteoblasts compared to other cell types in epiphysis? In situ measurement of Igf-1 mRNA abundance in different cell populations in epiphysis is needed to determine whether local Igf-1 expression in other cell types compensate the loss of osteoblastic Igf-1 in this skeletal site.

Response:  Please see response to reviewer #1 comment #1.

  1. In discussion, please add a paragraph discussing the possibility of the local contribution of Igf-1 expression in progenitor cells within the bone microenvironment to bone formation.  Deletion of Igf-1 using a Cre deleter strain targeting early progenitor cells such as Prx1-Cre may recapitulate the full phenotype of the global knockout mice.

Response:  Thank you for the suggestion.  We have added the potential contribution of IGF-I expression in osteoprogenitors as well as other cells in the bone marrow microenvironment in regulating skeletal phenotype. 

  1. Please check the references. Reference 30, 31 are not correctly cited and references 32-36 are missing.

Response:  We apologize for the errors and missing references. We have corrected these errors.  Thanks

Round 2

Reviewer 1 Report

Questions have been addressed.

Reviewer 3 Report

I have no further comments.